# A Delay Prediction Method for the Whole Process of Transit Flight

Zheng Zhao [1], Shicheng Feng [1,*], Meiwen Song [1] and Qizhao Liang [2]

1 Department of Civil Aviation, Nanjing University of Aeronautics and Astronautics, Nanjing 210001, China
2 Civil Aviation Central South Regional Air Traffic Administration, Guangzhou 510080, China
* Correspondence: fengshicheng@nuaa.edu.cn

**Abstract:** In order to strengthen the construction of smart airports and improve the ability of airport managers to identify, intervene and rescue delayed flights, this paper proposes a delay prediction method for the whole process of transit flights through the basic steps of node time and link time prediction and delayed flight identification. By designing the key node time prediction model (ML-DM), the method predicts the important guaranteed node time involved in the process of flight departure from the outstation to the departure from the current station. By constructing the imbalance data classification model, the delayed flight is identified at each predicted guarantee node. The experimental results for a busy airport show that this prediction method can achieve a maximum recognition rate of 96.5% for delayed flights.

**Keywords:** transit flight; delay prediction; ML-DM model; imbalanced data classification model

## 1. Introduction

The smart airport concept is the future of airport operations, and it may dramatically change the industry towards modern technology adaptation [1]. As a result of the fourth industrial revolution, the smart airport concept has been evolving all over the world, and it will eliminate the drawbacks of the conventional airport system. According to Bouyakoub et al., Airport 4.0 is a concept that leverages big data and open data to enhance its own innovation [2]. In its 2019 information circular, "Smart Airport Development Research and Practice Report" [3], the Department of Airports of the Civil Aviation Administration of China (CAAC) summarises the future development goals and trends of airports through research on and the analysis of international smart airport development and practice in a number of countries, including the US, Europe, the EU, Japan, Singapore and Dubai, where trends in airport development state that "Managers will be able to perceive aircraft security warnings in a timely manner; flight delays will be reduced to a minimum; airport resources will be optimally allocated and resource utilisation will be extremely high". The above requires airports to accurately identify delayed flights and to take reasonable and effective interventions for potentially delayed flights upon receipt of an alert. The scientific issues involved include the prediction of delayed flights and the implementation of delayed flight rescue measures.

Most of the existing papers on flight delay prediction have focused on both influence factor extraction and prediction models. Khaksar et al. analysed flight delays in the U.S. airline network based on machine learning, and the results of the study showed that visibility, wind, and departure time have large impacts on flight delays [4]. Truong et al. used two methods, decision trees and Bayesian inference, to predict the probability of flight delay events and constructed several flight delay prediction models from flight data from different sources; they then described the airport-related related important factors and their impacts on flight punctuality [5]. Wu et al. constructed a flight delay prediction model based on deep SE-DenseNet based on the fusion of flight information, related

airport delay information and weather information data, and the experimental results showed that the prediction accuracy improved by about 1.8% after information fusion compared with considering only flight attributes [6]. Esmaeilzadeh et al. analysed, based on support vector machine, the main factors that cause flight delays, and the analysis showed that delayed delays, slip-out delays, and ground waiting procedures had the greatest impacts on flight delays [7]. Choi et al. constructed a flight delay prediction model under severe weather conditions based on data mining and supervised machine learning algorithms and compared the prediction results of several algorithms, and the results showed that random forest had the highest prediction accuracy [8]. Ye et al. constructed four prediction models based on multiple linear regression, support vector machine, extreme random tree and LightGBM, and the results showed that the LightGBM model had the best prediction results [9]. Thiagarajan et al. constructed six departure flight delay prediction models based on machine learning algorithms, and the experimental results showed that the model constructed based on the gradient-boosting algorithm had the highest prediction accuracy [10]. Qu et al. constructed two flight delay prediction models based on deep convolutional neural networks, DCNN and SE-DenseNet, and achieved 92.1% and 93.19% prediction accuracy, respectively [11]. Yazdi et al. constructed three flight delay prediction models, SDA-LM, SAE-LM and SDA, based on deep learning. Experimental results on balanced and unbalanced datasets show that the SDA-LM model has the best prediction effect, with a prediction accuracy of up to 96%, and the prediction effect on balanced datasets is better than that on unbalanced datasets [12]. Ding et al. constructed a multiclassification prediction model for flight delays based on LightGBM and imbalanced the data by few oversampling techniques with TomekLink, and their prediction accuracy reached more than 90% [13]. Basturk et al. constructed a flight arrival time prediction model based on random forest and deep neural network considering flight, track and weather information, and the results showed that the prediction error of both could be controlled within 6 min [14]. Khan et al. proposed a hierarchical integrated machine learning model and used different machine learning algorithms and sampling methods to analyse and validate the proposed model using Hong Kong International Airport as the research object. The results showed that the model constructed based on the SMOTETomek sampling technique and the hyp-free CPCLS machine learning algorithm worked best [15]. Jiang Yu et al. constructed a departure flight delay prediction model based on a spatio-temporal graph convolutional neural network, and the experimental results showed that the model could significantly improve the accuracy of flight delay prediction compared with the historical averaging method, long and short-term memory recurrent neural network, and stacked self-encoder [16]. Roger et al. proposed a departure flight delay prediction method based on XGBoost and Logistic, which focuses on the effect of sparse data on the flight delay prediction model. The experimental results show that the method can significantly improve the prediction of the model on sparse data sets [17].

Existing studies have achieved good results in the extraction of flight delay influencing factors and the construction of prediction models, but the prediction of delayed flights mainly stays in the static stage and provides limited decision support for how to save the delayed flights after identification. In actual operation, predicting and identifying delayed flights is only a means to an end: taking effective measures to avoid delays after they are identified is the goal. Compared with the originating flights, the transit flights have more guaranteed links and require more coordination capability among airports, airlines and ATC. Therefore, considering the whole process of flight transit, the flight transit process is divided into four stages: approach, taxi-in, turnaround and taxi-out. The first three intervention stages are used to predict and identify delays in turn, and by predicting the time spent in the three stages, the possible locations of delays are located, and corresponding intervention measures are taken for different locations of delays. The ultimate goal is to identify delayed flights and provide decision support on what delay intervention measures to take, thereby avoiding flight delays.

This paper consists of four chapters. Chapter 1 introduces the guaranteed process of the transit flight and the key nodes involved and defines the scope of the research on delay prediction in this paper. Chapter 2 introduces the key issues in the transit flight delay prediction method, including the definition of delay thresholds, the complete process of prediction, and the description of the involved models. Chapter 3 verifies the effectiveness of the proposed delay prediction method for transit flights using a busy airport in China as the research object. Chapter 4 summarizes the contents of this paper.

## 2. Key Nodes and Guarantee Process of Transit Flight

In order to achieve effective monitoring and management of the flight guarantee links, the Airport Collaborative Decision Making (ACDM) system collects and configures 45 flight guarantee nodes, including landing, in-block, off-block and take-off [18]. In this paper, we focus on the overall process and some of the key nodes involved, and on the basis of ensuring the integrity of the process, we discard some nodes and add the nodes required in this paper to build the transit flight guarantee process, as shown in Figure 1.

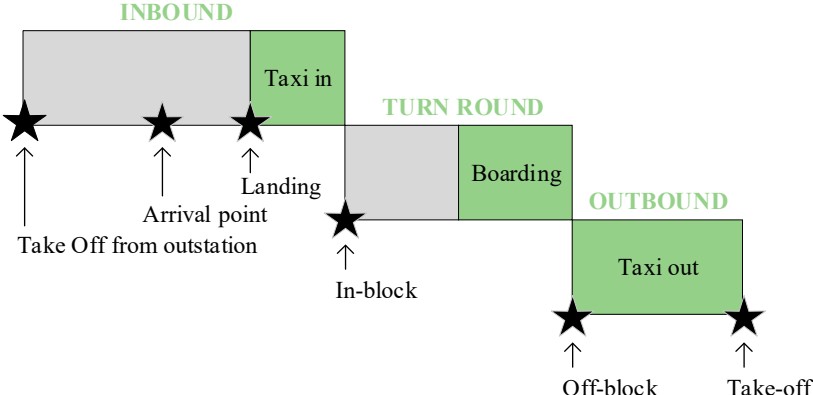

**Figure 1.** Transit flight guarantee process.

In order to ensure that the flight ground guarantee process can be completed on time and efficiently, some guaranteed equipment needs to be in place before the in-block time [19]. Therefore, by accurately predicting the in-block time of the flight and even the landing time, the airport can dynamically adjust the time when the flight guarantee resources are in place and improve the efficiency of guaranteed resources utilization, thus avoiding flight delays caused by the availability of guaranteed resources to a certain extent. In addition, in the turnaround process, the start, elapsed time and end of different guaranteed operations are uncertain and may cause flight delays due to partial operational delays. Therefore, by accurately predicting the off-block time at the in-block time, the airport can intervene to ensure that the process can be completed on time and avoid flight delays due to delays in the process to a certain extent. Therefore, the guaranteed node moment prediction and flight delay prediction are performed sequentially at different stages from the flight passing through the arrival point to departure, which can provide more options for the adoption of intervention measures for delayed flights and also provide a buffer time for the implementation of the measures.

For the delay prediction of the whole process of transit flights, the delayed flight is identified when the flight passes the arrival point and the landing time, and at the same time, the estimated landing time and the estimated in-block time of the flight are obtained by predicting the approach time and taxi-in time respectively, to provide decision support for the airport to intervene the time when guarantee resources are in place; The delay identification is performed at the time of the in-block, and at the same time, the turnaround time of the flight is predicted to obtain the estimated off-block time, which provides decision support for the airport to intervene in the guarantee operation between the in-block and the off-block.

## 3. Design of Delay Prediction Method for Transit Flight

This chapter introduces the basis for defining flight delay thresholds and the general process of prediction and provides a detailed description of the specific models and methods involved in the prediction process, including the design and implementation process of the key node time prediction model, the statistical method of empirical taxi time and the construction method of the imbalanced data classification model, as well as the machine learning algorithms involved.

### 3.1. Flight Delay Threshold

The Civil Aviation Administration of China's standard for the normal determination of departing flights is: "A flight is normal if it takes off within the standard ground taxi time after the planned departure time and no abnormal conditions such as return or standby occur" [20]. Accordingly, in this paper, the difference between the actual take-off time (ATOT) and the scheduled off-block time (SOBT) and the standard taxi time (STT) of a flight is used as the threshold for determining whether a flight is delayed, and the calculation method is shown in Equation (1):

$$D = \begin{cases} 0 & \text{ATOT} - \text{SOBT} - \text{STT} \leq 0 \\ 1 & \text{ATOT} - \text{SOBT} - \text{STT} > 0 \end{cases} \tag{1}$$

where $D = 0$ represents normal flight, $D = 1$ represents abnormal flight, ATOT represents actual take off time, SOBT represents scheduled off-block time, and STT represents the standard taxi time.

### 3.2. Flight Delay Prediction Process

In this paper, delay prediction are phased for transit flight to provide airports with more options for interventions and more time to implement the measures. The delay prediction process for transit flights is divided into three phases: Phase 1 takes the flight passing the arrival point time as the starting time; Phase 2 takes the flight landing time as the starting time; and Phase 3 takes the in-block time as the starting time. The prediction process of each phase is divided into two steps: node time and duration prediction and delayed flight identification, and the overall prediction process is shown in Figure 2.

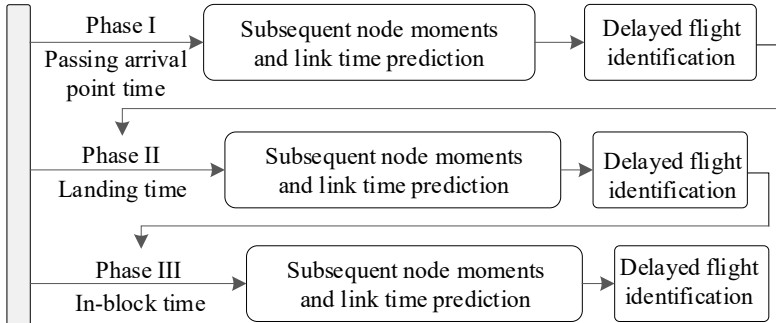

**Figure 2.** Flight delay prediction process.

Taking the first phase as an example, the flight delay prediction process and steps are explained in detail, as shown in Figure 3.

Step 1: Through the regression prediction and statistical mining methods included in the constructed key node time prediction model, the estimated time of each link in the flight approach process is obtained, and the estimated time of each subsequent node is projected based on the approach process with the time of passing through the arrival point as the starting point. The predicted time of the link provides the feature input for the delayed flight identification model, and the predicted time of the node provides support for the delayed flight intervention.

Step 2: Constructing a feature set based on the extracted influencing factors related to flight delays and the expected duration of the link obtained in the previous sequence, extracting data according to the feature set and identifying delayed flights on the basis of balancing the data by means of an imbalanced data classification model.

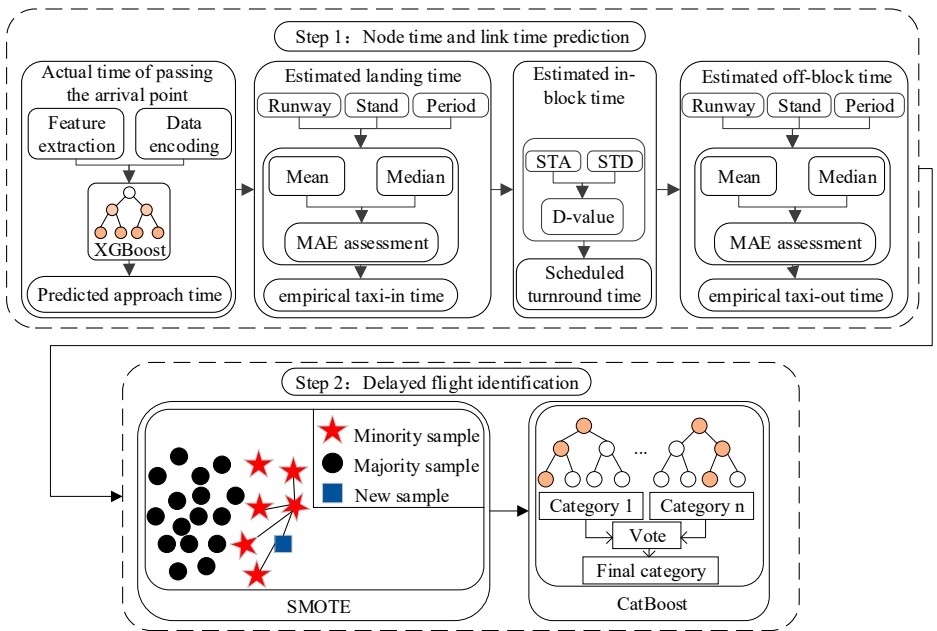

**Figure 3.** First phase prediction process.

3.2.1. Key Nodes Time Prediction Model

The key node time prediction model is constructed based on machine learning and data mining (ML-DM). The purpose of designing this model is to obtain the estimated time spent at each link and the estimated time at each node through the updated iterative method of the flight transit process, which is included in all three phases of transit flight delay prediction.

The model is divided into two parts: The first part predicts the duration by constructing a machine learning model as the feature input for the imbalanced data classification model, and it extrapolates the estimated time of the subsequent guarantee node based on the starting point time to provide decision support for the adoption of delayed flight intervention measures; the second part uses the empirical taxi time statistics designed based on data mining technology to obtain the corresponding link time to provide feature input for the imbalanced data classification model; then, it extrapolates the estimated time of the subsequent guarantee node based on the starting point time and the prediction result of the first part to provide support for the adoption of delayed flight intervention measures. The three phases of the transit flight delay prediction process are similar, and the differences are mainly reflected in the number of research objects, as shown in Figure 4, and the meanings of the acronyms in the figure are shown in Table 1.

The implementation process of phase 1 in Figure 4 is used as an example for detailed explanation, and the whole process is divided into four steps.

Step 1: Obtain the Estimated landing time (*ELDT*) by predicting the approach time (*PAT*) at the time when the flight actually passes through the arrival point (*APAT*), calculated as shown in Equation (2).

$$APAT \; + \; PAT \; = \; ELDT \tag{2}$$

Step 2: The Estimated in-block time (*EIBT*) is obtained by extracting the empirical taxi-in time (*ETI*), which is calculated as shown in Equation (3).

$$ELDT \; + \; ETI \; = \; EIBT \tag{3}$$

Step 3: Obtain the Estimated off-block time (*EOBT*) by obtaining the Scheduled connection time (*SCT*), which is calculated as shown in Equation (4).

$$EIBT + SCT = EOBT \tag{4}$$

Step 4: The Estimated departure time (*EDT*) is obtained by extracting the empirical taxi out time (*ETO)*, which is calculated as shown in Equation (5).

$$EOBT + ETO = ETOT \tag{5}$$

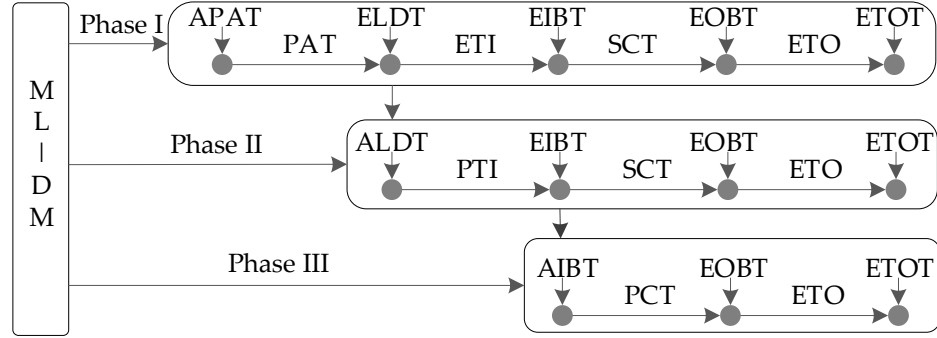

**Figure 4.** ML-DM model implementation process.

**Table 1.** Acronyms in the implementation process of ML-DM model.

| Acronyms | Definition |
|---|---|
| APAT | Actual passing the arrival point time |
| PAT | Predicted approach flight time |
| ELDT | Estimated landing time |
| ETI | Experience taxi-in time |
| EIBT | Estimated in-block time |
| SCT | Scheduled connection time |
| EOBT | Estimated off-block time |
| ETO | Experience taxi-out time |
| ETOT | Estimated take off time |
| ALDT | Actual landing time |
| PTI | Predicted taxi-in time |
| AIBT | Actual in-block time |
| PCT | Predicted connection time |

(1) Duration Prediction Method

The duration prediction includes three parts: approach time, taxi-in time and turnaround time. Each part of the prediction follows the following steps: pre-processing of airport operational data, qualitative analysis of influencing factors to extract candidate factor sets, quantitative analysis of influencing factors to filter and condense the final factor sets, feature extraction and calculation to build model input data sets, algorithm selection and hyperparameter adjustment to build prediction models, and evaluation of model prediction effects. Among them, the construction of the model input dataset is based on the final set of influencing factors for the corresponding feature extraction and the calculation of the data, while in machine learning prediction, the quality of the input dataset has a much greater impact on the model prediction effect than the selection of the prediction algorithm and the adjustment of the model hyperparameters, so the extraction of influencing factors is crucial. In this paper, the method and steps of duration prediction are shown in Figure 5, and the method and steps of influence factor extraction are shown in Figure 6.

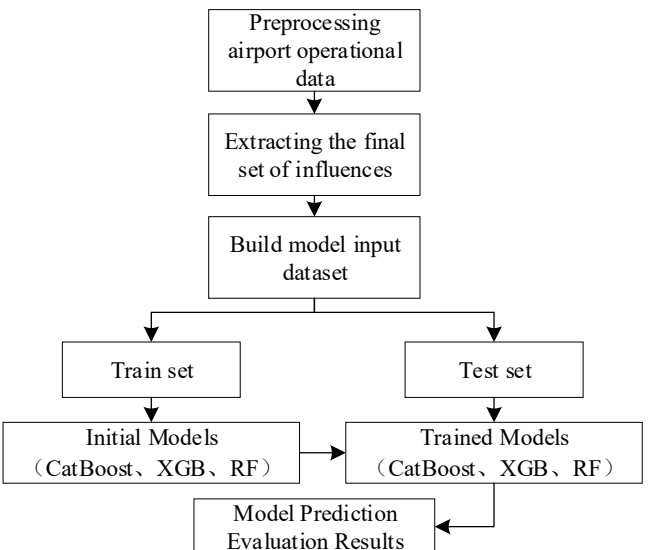

**Figure 5.** Methods and steps for duration prediction.

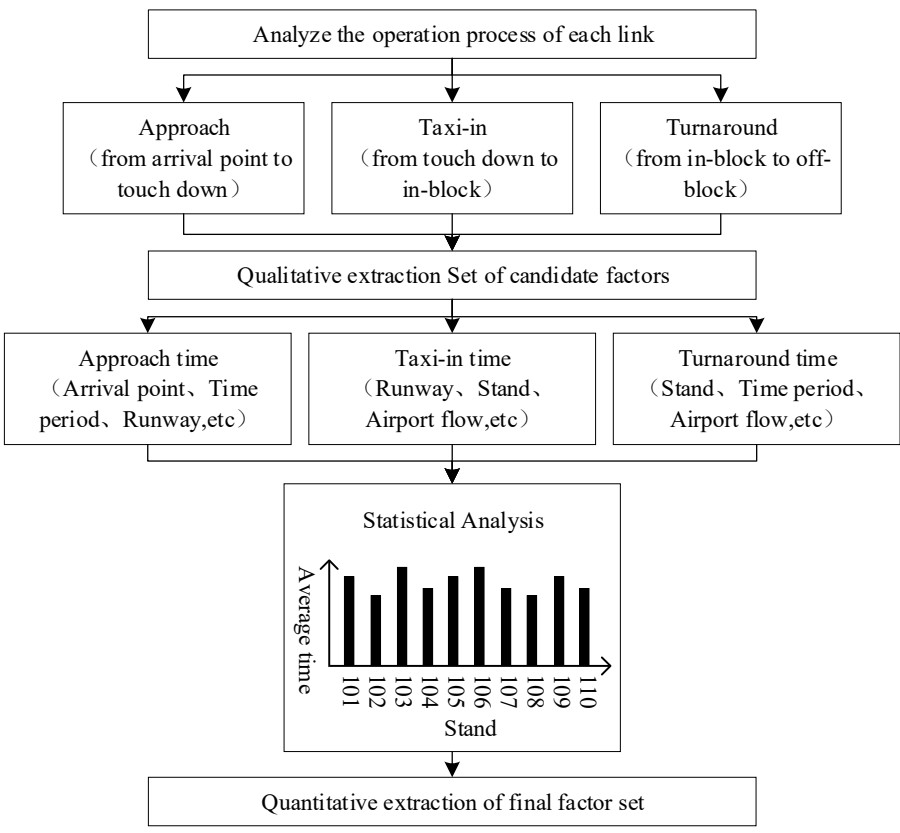

**Figure 6.** Influencing factors extraction method and steps.

(2) Statistical Method of Empirical Taxi Time

The empirical taxi time statistical method is included in the key node time prediction model, including the taxi-in and taxi-out time, and the statistics of the two are used to calculate the estimated time of subsequent nodes. The taxi-in time refers to the time spent from landing to the in-block of the flight, and the taxi-out time refers to the time spent from the off-block of the flight to take-off, both of which are calculated as follows.

$$T_{\text{Taxi-In}} = T_{\text{In-Block}} - T_{\text{Touch-Down}} \tag{6}$$

$$T_{\text{Taxi-Out}} = T_{Take\text{-}Off} - T_{Off\text{-}Block} \tag{7}$$

The statistical methods for taxi-in and taxi-out times are the same, and both follow the following process: mining the historical airport operation data, selecting the candidate values, statistically comparing MAE (Mean Absolute Error, MAE), and determining the final values. The specific method is illustrated as an example of taxi-in time statistics process. First, the data are divided into two groups, data mining group and result verification group; then, the data are aggregated and grouped by inbound runway, inbound stand and inbound time period based on the data mining group, and the mean and median of each group are counted; then, the obtained mean and median are brought into the result verification group to calculate MAE respectively; finally, the final value is determined based on the MAE comparison result. The specific process is shown in Figure 7.

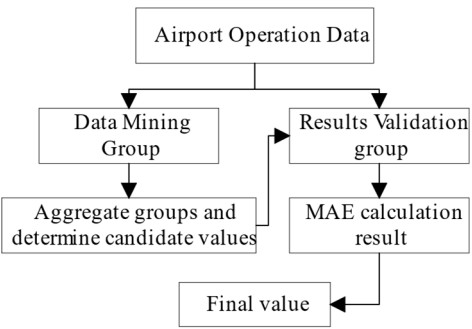

**Figure 7.** Process of empirical taxi time statistics.

3.2.2. Imbalanced Data Classification Model

The amount of data between normal and delayed flights in the transit flight data differs greatly, and in order to solve the problem of poor accuracy of the classification model that may be caused by sample imbalance, this paper introduces a classification model based on SMOTE algorithm (Synthetic Minority Oversampling Technique) and CatBoost algorithm (Categorical Boosting) to introduce an imbalanced data classification model, namely the SMOTE-CatBoost model.

SMOTE algorithm is an oversampling processing technique applied to imbalanced sample data proposed by Chawla et al. [21]. Unlike the simple copying sample mechanism of random oversampling, the SMOTE algorithm synthesizes a new sample between two minority class samples by linear interpolation, which can avoid the overfitting problem generated in random oversampling to some extent. The main steps of the algorithm are as follows.

(1) For each sample x in the minority class, calculate the distance between that point and the other sample points in the minority class to obtain the nearest k nearest neighbours.

(2) Set the sampling ratio to determine the sampling multiplier, for each randomly selected nearest neighbour x′.

(3) For each randomly selected nearest neighbour x′, a new sample is constructed separately from the original sample according to the following formula.

$$x_{new} = x + rand(0,1) \times (x' - x) \tag{8}$$

CatBoost algorithm is a GBDT framework with symmetric decision tree as the base learner and supports category-based variables. The CatBoost algorithm optimises the constructed model by constructing a learner with reduced loss along the steepest direction of the gradient at each iteration step. The algorithm model can be defined as [22].

$$F(x,\omega) = \sum_{t=0}^{T} \alpha_t h_t(x,\omega) = \sum_{t=0}^{T} f_t(x,\omega_t) \tag{9}$$

In the formula: $L(\cdot)$ is the output of the whole decision tree, x is the input of the sample, $\omega$ is the parameter of the whole decision tree, $\alpha_t$ is the weight of the t-th tree, T is the tree tree, $h_t(\cdot)$ is the output of the t-th decision tree, $\omega_t$ is the parameter of the t-th decision tree, $f_t(\cdot)$ is the output of the t-th decision tree after weighting.

The parameters of the optimal model were obtained by minimizing the loss function as [22].

$$(\alpha_t, \omega_t) = \arg\min \sum_{i=0}^{N} L(y_i, F(x_i, \omega)) \tag{10}$$

In the formula: $L(\cdot)$ is the loss function, $y_i$ is the actual output of sample i, $x_i$ is the input of sample i, and $N$ is the sample size.

The CatBoost algorithm uses ordered boosting to obtain an unbiased estimate of the gradient, which mitigates the effect of the bias in the gradient estimation error and improves the generalization ability of the model. In order to solve the conditional bias problem arising when the traditional GBDT algorithm uses the label mean as the node splitting criterion, the CatBoost algorithm adds prior terms and weight coefficients to solve the bias problem by reducing the influence of noise and low frequency category data on the data distribution [23].

$$\hat{x}_k^i = \frac{\sum\limits_{j=1}^{N} I_{\{x_j^i = x_k^i\}} y_j + ap}{\sum\limits_{j=1}^{N} I_{\{x_j^i = x_k^i\}} + a} \tag{11}$$

In the formula: $x_k^i$ is the i-th category feature of the k-th training sample, $\hat{x}_k^i$ is its average; $y_j$ is the label of the j-th sample; $I$ is the indicator function, i.e., go to 1 when the two quantities in brackets are equal, otherwise take 0; $p$ is the prior term; a is the weighting factor.

The CatBoost algorithm solves the problem of gradient bias and prediction bias, which reduces the occurrence of overfitting and improves the generalisation ability of the algorithm while increasing its accuracy.

The SMOTE-CatBoost model prediction process is shown in Figure 8. Firstly, the pre-processed dataset is divided into train and test set according to a certain ratio; then, the SMOTE algorithm is used to generate minority category samples in the training set and train CatBoost based on the balanced dataset; finally, the trained CatBoost classifier is used to predict the test set and the prediction effect of the model is evaluated.

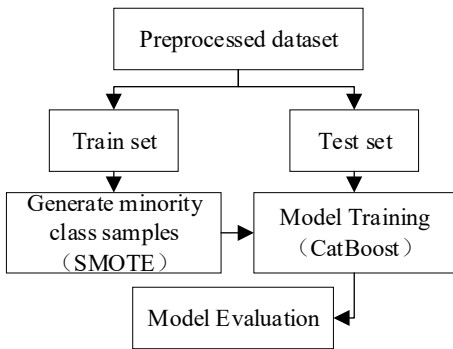

**Figure 8.** SMOTE-CatBoost model prediction process.

## 4. Case Study

In order to verify that the proposed method of transit flight delay prediction can be applied to actual airport operations, this section conducts tests based on actual airport operational data to verify the effectiveness of the method in this paper. The experimental algorithms were written in Python 3.7.0.

*4.1. Data Preprocessing*

The experimental data are obtained from the actual operational data of a busy airport in China. Firstly, the transit flights are filtered according to the planned turnaround time. Then, some of the data that do not conform to the time logic and business logic are adjusted or deleted; the missing values that cannot be filled are deleted, such as time type data; the missing values that can be filled, such as runway and stand, are filled according to the airport operation characteristics. Finally, based on the filtered data set, the empirical taxi-in and taxi-out times are counted based on the method described above and filled into the data set, and the sample data obtained is shown in Table 2.

**Table 2.** Sample data.

| Name | Content | Name | Content | Name | Content |
|---|---|---|---|---|---|
| Airline | KN | STA | 1 february 2021 11:50 | Outbound stand | 104 |
| Aircraft type | 738 | ATA | 1 february 2021 11:10 | Outbound runway | 35R |
| Arrival point | A3 | In-block | 1 february 2021 11:20 | Empirical taxi-in time | 6.50 |
| Time of passing arrival point | 2021/02/01 10:39 | Off-block | 1 february 2021 12:43 | Scheduled turnaround time | 65 |
| Inbound runway | 01L | STD | 1 february 2021 12:55 | Empirical taxi-out time | 16.91 |
| Inbound stand | 104 | ATD | 1 february 2021 12:57 | | |

In the statistics of the number of delayed flights in the dataset, it was found that among about 25,000 data, the data of delayed flights totalled 4710, accounting for less than 20%. Generally, in classification tasks, when the ratio of training samples of different categories is significantly larger than 1:1 it can be classified as a sample imbalanced problem, while in the dataset studied in this paper, the ratio of normal flights to delayed flights is larger than 4:1, which is a sample imbalanced problem.

*4.2. Prediction Modeling*

4.2.1. Duration Prediction Model

In the construction of the approach time influence factor set, the influence of the company and aircraft on the approach flight is reflected by the airline company and aircraft type, the influence of the approach status on the approach flight is reflected by the approach time, approach speed, arrival point and approach altitude, the influence of the airspace busyness on the approach flight is reflected by the number of incoming flights and the number of departing flights, and the influence of the approach route on the approach flight is reflected by the landing runway.

In the construction of the set of factors influencing taxi-in time, the influence of company and aircraft on taxi-in is reflected by the airline company and aircraft type, the influence of surface busyness on taxi-in is reflected by the number of aircraft taxiing at the same time and landing time period, the influence of taxiing route on taxi-in is reflected by landing runways, inbound stand and the number of hot spots passing through, the influence of airport-specific layout on taxi-in is reflected by whether crossing the runway and whether cross taxi-in.

In terms of the construction of the set of factors influencing the turnaround time, the influence of the company and aircraft on flight guarantee is reflected by the airline and aircraft type; the airport guarantee efficiency in the current period is reflected by the average turnaround time of the first 15 and 30 min flights; the influence of time urgency on turnaround guarantee is reflected by the planned turnaround time and the length of inbound delays; and the influence of demand on turnaround guarantee is reflected by in-block time period, the number of flights landing in the first 15 and 30 min. The set of influencing factors of the three links is shown in Table 3.

**Table 3.** The set of influencing factors of the three links.

| Number | Approach | Taxi-in | Turnaround |
|---|---|---|---|
| 1 | Airline | Airline | Airline |
| 2 | Aircraft type | Aircraft type | Aircraft type |
| 3 | Arrival point | Landing time period | In-block time period |
| 4 | approach altitude | Landing runway | length of inbound delay |
| 5 | approach speed | Landing Stand | Planed turnaround time |
| 6 | Inbound runway | Number of Hot Spots | Average turnaround time in the first 15 min |
| 7 | Inbound time period | Number of aircraft taxiing on the surface at the same time | Average turnaround time in the first 30 min |
| 8 | Number of inbound flights | Whether crossing the runway | Number of landing flights in the first 15 min |
| 9 | Number of outbound flights | Whether cross taxi-in | Number of landing flights in the first 30 min |

Based on the constructed set of influencing factors, the individual features used to input the model were extracted and calculated from the pre-processed data, and the text-based data were coded and processed to obtain the final input data for the three links duration prediction model. Taking the taxi-in time as an example, the model input data is shown in Table 4.

**Table 4.** Taxi-in time prediction model inputs.

| Month | 1 | 1 | Inbound cumulative flow | 5 | 7 |
|---|---|---|---|---|---|
| Hour | 23 | 0 | Outbound instantaneous flow | 1 | 5 |
| Airline | 0 | 1 | Outbound cumulative flow | 3 | 8 |
| Aircraft type | 1 | 2 | Whether crossing the runway | 1 | 0 |
| Runway | 0 | 1 | Whether crossing the runway | 0 | 1 |
| Stand | 5 | 12 | Number of Hot Spots | 0 | 2 |
| Inbound instantaneous flow | 2 | 3 | | | |

Approach time, Taxi-in time and Taxi-out time were modelled separately and the key hyperparameters of the algorithm were tuned using a grid search to obtain the best values for the key hyperparameters of the three models built on the CatBoost algorithm as shown in Table 5.

**Table 5.** Optimal values of key hyperparameters of the three phases model.

| | Iterations | Max_depth | Subsample | Learning_rate | L2_leaf_REG |
|---|---|---|---|---|---|
| Approach time | 800 | 5 | 1 | 0.15 | 4 |
| Taxi-in time | 600 | 4 | 0.9 | 0.12 | 4 |
| Taxi-out time | 1000 | 6 | 0.9 | 0.18 | 3 |

### 4.2.2. Flight Delay Prediction Model

In the construction of imbalanced data classification model feature set, the impact of company, aircraft, inbound and outbound procedure on flights is reflected by airline company, aircraft type, arrival point, runway and stand; the impact of traffic on flights is reflected by time period and duration in each links obtained based on ML-DM model. The third phase reflects the impact of inbound delay on subsequent flights by the length of inbound delay. The final feature sets of the three phases are shown in Table 6.

**Table 6.** Feature sets of the three phases of the imbalanced data classification model.

| Number | Phase 1 | Phase 2 | Phase 3 |
|--------|---------|---------|---------|
| 1 | Airline | Airline | Airline |
| 2 | Aircraft type | Aircraft type | Aircraft type |
| 3 | Arrival point | Landing time | In-block time |
| 4 | Time of passing arrival point | Inbound runway | length of inbound delay |
| 5 | Inbound runway | Inbound stand | Inbound runway |
| 6 | Inbound stand | Outbound stand | Inbound stand |
| 7 | Outbound stand | Outbound runway | Outbound stand |
| 8 | Outbound runway | Predicted taxi-in time | Outbound runway |
| 9 | Predicted approach time | Scheduled turnaround time | Predicted turnaround time |
| 10 | Empirical taxi-in time | Empirical taxi-out time | Empirical taxi-out time |
| 11 | Scheduled turnaround time | | |
| 12 | Empirical taxi-out time | | |

Based on the constructed feature set, each feature data used for the input model is extracted from the preprocessed data and the results obtained from the ML-DM model, and the text class data is processed using the same data coding method as above to finally obtain the input data for the imbalanced data classification model in three phases. Taking the second phase as an example, the input data of the model are shown in Table 7.

**Table 7.** Input data for the second phase imbalance data classification model.

| Name | Content | Name | Content |
|------|---------|------|---------|
| Airline | 0 | Outbound stand | 12 |
| Aircraft type | 1 | Outbound runway | 1 |
| Landing time | 11 | Predicted approach time | 8.0 min |
| Inbound runway | 0 | Scheduled turnaround time | 65 min |
| Inbound stand | 12 | Empirical taxi-out time | 15.45 min |

After balancing the dataset using the SMOTE algorithm, the established CatBoost classification model is tuned using grid search based on the balancing processed data. In order to increase the applicability of the model, the optimal values of the key parameters of the classification model in the three phases were consistent, and the optimal values of the key parameters of the model were obtained on the basis of ensuring the prediction effect in each stage as shown in Table 8.

**Table 8.** Optimal values of key hyperparameters of CatBoost model.

| | Iterations | Depth | Subsample | Learning_rate | L2_leaf_reg |
|------|------------|-------|-----------|---------------|-------------|
| Value | 700 | 5 | 0.77 | 0.10 | 3 |

*4.3. Analysis of Experimental Results*

4.3.1. Empirical Time Statistics Results

Using the empirical taxi time statistics, the MAE statistics obtained based on the pre-processed data set with median and mean values as candidates, respectively, are shown in Table 9. From Table 9, it can be seen that the MAE obtained by using the median for both taxi-in time and taxi-out time is significantly smaller than the mean value, and the error of MAE compared to the mean value of both is within 18%, and this accuracy can meet the airport operation requirements, so the median is used as the final value of empirical taxiing time.

**Table 9.** MAE Statistical Results.

| | Taxi-in Time | | Taxi-out Time | |
|---|---|---|---|---|
| Candidate Value | Mean | Median | Mean | Median |
| MAE | 3.93 min | 1.95 min | 5.39 min | 4.2 min |

### 4.3.2. Duration Prediction Results

The prediction results of taxi-in time, turnaround time and taxi-out time based on the tuned CatBoost model are shown in Table 10. From Table 10, the predicted MAE is 1.98 min with the mean value of approach time of 23.8 min, and the prediction accuracy within the error range of $\pm 5$ min reaches 92.8%; the predicted MAE is 1.37 min with the mean value of taxi-in time of 12.1 min, and the prediction accuracy within the error range of $\pm 3$ min reaches 92.5%. The estimated landing time and the estimated in-block time of the flight obtained under this accuracy condition can provide decision support for the airport to intervene in the possible delayed flight by scheduling the availability of guarantee resources. The predicted MAE is 4.66 min with a mean connection time of 69.7 min, and the prediction accuracy within the error range of $\pm 15$ min reaches 97.6%. This accuracy can provide more reliable input features for the imbalance data classification model.

**Table 10.** Prediction results of approach time, taxi-in time and turnaround time.

| | Mean (min) | MAE (min) | | | | Accuracy (%) | | |
|---|---|---|---|---|---|---|---|---|
| | | CatB | XGB | RF | | CatB | XGB | RF |
| Approach time | 23.8 | 1.98 | 2.08 | 2.30 | $\pm 5$ min | 92.8 | 92.1 | 88.5 |
| Taxi-in time | 12.1 | 1.37 | 1.44 | 1.63 | $\pm 3$ min | 92.5 | 91.5 | 87.2 |
| Turnaround time | 69.7 | 4.66 | 4.75 | 5.08 | $\pm 15$ min | 97.6 | 96.5 | 92.8 |

### 4.3.3. Flight Delay Prediction Results

The predicted MAE for the remaining time spent on a transit flight and recognition rate of delayed flights obtained based on the ML-DM model are shown in Figure 9. As can be seen from Figure 7, there is not much difference in the MAE of prediction results and the recognition rate of delayed flights in both phase 1 and 2, and the MAE of phase 2 decreases less than 7% and the recognition rate increases less than 3% compared with phase 1. In phase 3, the MAE of predicted time decreases from 22.08 min in the previous phase to 7.69 min, with a decrease of 66.3%; while the recognition rate of delayed flights increases from 52.5% in the previous phase to 72.1%, with an increase of 37.3%, which is a significant improvement in the prediction effect, which is due to the fact that phase 3 replaces the planned turnaround time used in the previous two phase with the predicted turnaround time. The reason for this is that phase 3 is closer to the actual turnaround time after replacing the planned turnaround time with predicted turnaround time. However, the 72.1% delayed flight recognition rate obviously does not meet the actual operational requirements.

To further improve the recognition rate of delayed flights, an imbalanced data classification model was introduced based on the ML-DM model. For the evaluation of classification results, precision rate, recall rate and ROC (receiver operating characteristic) curve are selected. The precision rate represents the percentage of correct predictions among the samples with positive predictions, which in this paper represents the recognition rate of normal flights; the recall rate represents the percentage of correct predictions among the samples with negative predictions, which in this paper represents the recognition rate of delayed flights; the ROC curve can visually evaluate the model as a whole.

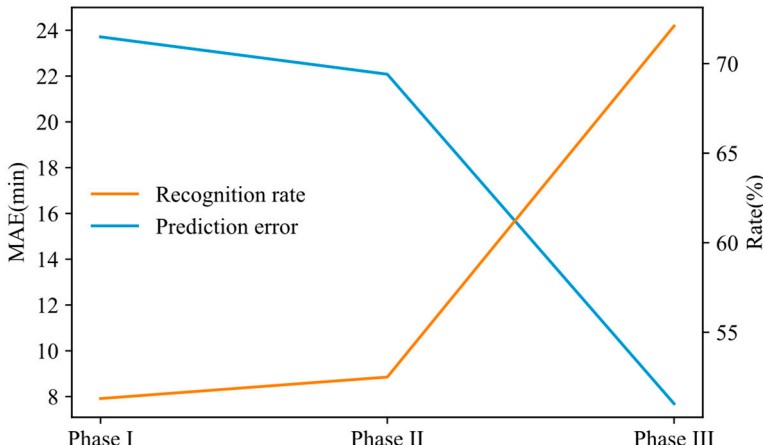

**Figure 9.** Remaining time use prediction error and delayed flight identification rate.

The calculation results of the precision and recall rates of the three phases of flight transit are shown in Table 11, the flight transit process is shown in Figure 10, and the ROC curve is shown in Figure 11. Combining Table 11 and Figure 10, it can be seen that at the passing arrival point time of flight transit phase 1, the recognition rates of the model for normal and delayed flights are 88.9% and 83.6%, respectively, neither of which reaches 90%. However, at the landing time of phase 2, the recognition rate of the model for normal and delayed flights increases to 93.0% and 90.4%, respectively, which has reached the high level of the current papers of the same type. By the in-block time of phase 3, the model achieved 96.3% and 96.5% recognition rate for normal and delayed flights, respectively, and exceeded the recognition rate of the same type of paper by 4% for delayed flights. It can also be seen visually from Figure 11 that the ROC curve gradually moves to the upper left corner from phase 1 to phase 3, and the AUC value (the area under the curve, which takes values between 0 and 1) gradually increases. Although the recognition rate of delayed flights in the first phase is relatively low, the unrecognized flights can be further identified in the subsequent phases, so on the whole this relatively independent and progressive identification method can avoid missing delayed flights to the greatest extent, and at the same time the phased intervention measures can provide a larger buffer space for the airport.

**Table 11.** Results of precision and recall rates in three phrases.

|  | Precision Rate | Recall Rate |
|---|---|---|
| Time of passing arrival point | 88.9% | 83.6% |
| Landing time | 93.0% | 90.4% |
| In-block time | 96.3% | 96.5% |

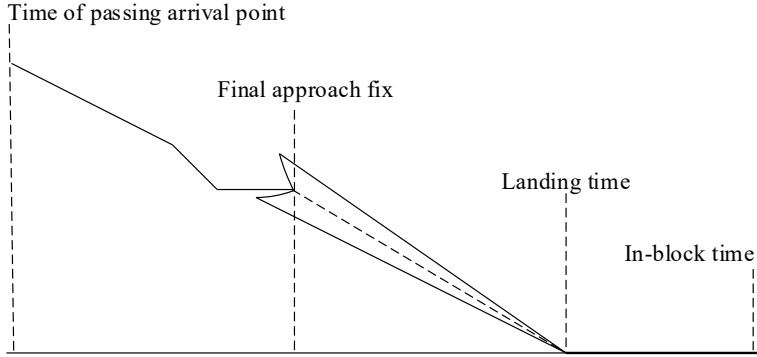

**Figure 10.** Schematic diagram of the flight approach process.

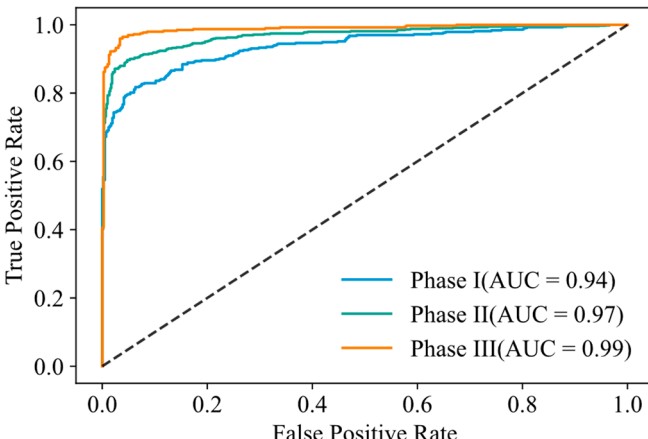

**Figure 11.** ROC graphs for the three phrases.

Figure 10 shows the comparison of the recognition rate of delayed flights using only the XGB-DM model and using both the XGB-DM model and the imbalanced data classification model. As can be seen from the Figure 12, compared with the delay prediction using only the XGB-DM model, the inclusion of the imbalanced data classification model results in a greater increase in the recognition rate of delayed flights in all three phases, from a low of 51.3% to 83.6% in phase 1 and from a high of 72.1% to 96.5% in phase 3.

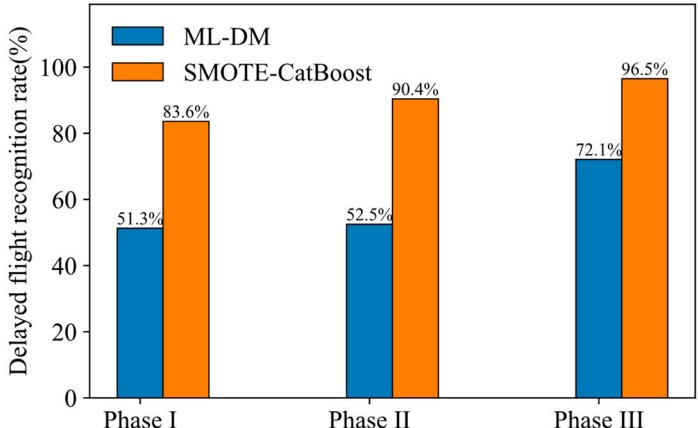

**Figure 12.** Delayed flight recognition rate comparison chart.

Table 12 shows the comparison between the direct flight delay prediction, i.e., the features corresponding to the duration in the imbalanced data classification model are removed, and the delay prediction based on the duration. As can be seen from Table 12, the prediction results based on the duration are higher than the direct prediction in terms of precision and recall in all three phases of the transit flight, which further confirms the effectiveness of the critical node moment prediction model, and that the predicted duration can reflect the surface operating conditions to a certain extent and thus can be used as input features for the imbalanced data classification model to improve the prediction effect.

**Table 12.** Comparison of prediction results between the two methods.

|  | Precision | | Recall | |
| --- | --- | --- | --- | --- |
|  | Method 1 | Method 2 | Method 1 | Method 2 |
| Phase I | 87.6% | 88.9% | 81.5% | 83.6% |
| Phase II | 91.4% | 93.0% | 87.8% | 90.4% |
| Phase III | 93.8% | 96.3% | 94.2% | 96.5% |

Table 13 shows the comparison of the nodes and model prediction accuracy of this paper for judging flight delays with existing research results. The existing predictions of flight delays for departing flights are all judged by the time of off-block, while this paper chooses the time of departure as the judgment criterion and makes multiple predictions in the whole process of flight transit. This is, firstly, more conducive to the managers' judgment and early warning of delays in the whole process of flight operation; secondly, multiple predictions and warnings in the process of flight transit are beneficial to the recovery of delays; thirdly, it is more in line with the requirements of the Civil Aviation Administration of China (CAAC) on the statistical standard of departing flights' normalcy. In terms of prediction accuracy, the method described in this paper achieves 86.3% in phase 1, 91.8% in phase 2, and 96.3% in phase 3, which is an improvement of more than 3% compared with the existing papers that use the time of off-block as the delay judgment criterion.

**Table 13.** Comparison between this paper and existing studies.

| Author | Node Used to Determine Flight Delays | | Accuracy |
| --- | --- | --- | --- |
| | **Off-Block Time** | **Take-Off Time** | |
| Khaksar et al. [3] (2017) | √ | × | 77.01% |
| Thiagarajan et al. [9] (2017) | √ | × | 86.48% |
| Wu Renbiao et al. [5] (2019) | √ | × | 92.39% |
| Ye et al. [8] (2020) | √ | × | 86.55% |
| Qu et al. [10] (2020) | √ | × | 93.19% |
| Ding Jianli et al. [11] (2021) | √ | × | 90.30% |
| Roger et al. [14] (2022) | √ | × | 92.90% |
| This paper | × | √ | Phase I: 86.3% Phase II: 91.8% Phase III: 96.3% |

## 5. Discussion

This paper addresses the problem that the current flight departure delay prediction method is difficult to provide decision support for saving delayed flights, and designs a phased delay prediction method for the whole process of transit flights, The main results are as follows. A machine learning-based key nodes time prediction model was constructed for the flight landing time, the in-block time and the off-block time. The model prediction results showed that the CatBoost algorithm outperformed XGBoost and Random Forest in terms of prediction error and prediction accuracy. The prediction accuracy of the three links reached 92.8%, 92.5% and 97.6% respectively using the CatBoost algorithm. A delayed flight prediction identification method based on the output of the key nodes time prediction model and machine learning was designed to perform flight delay prediction identification at the arrival point moment, the landing moment and the in-block moment respectively in sequence during the flight approach. Using the constructed imbalance data classification model combining SMOTE and CatBoost algorithm, the recognition rate of delayed flights in the three phases reached 83.6%, 90.4% and 96.5% respectively. The designed flight delay prediction method consists of a key nodes time prediction model and an imbalance data classification model, and stage predictions are made during the flight approach process. On the basis of ensuring that the accuracy of delayed flight prediction meets or exceeds existing research, the key nodes time prediction model is used to locate the possible locations of flight delays and provide decision support for the adoption of corresponding intervention measures, while the stage predictions also provide more time for the implementation of intervention measures.

The focus of this paper is on how to predict delayed flights while providing decision support to save them, therefore only the process of a transit flight from passing through the arrival point to departure is selected. The next phase of the study considers extending this process to include flights from outstation departures to current station departures, while

expanding the applicability of this method beyond transiting flights. In addition, weather was considered as a factor in the design of the delay prediction method, but as reliable weather data is not currently available, the effect of weather is not reflected in the paper.

**Author Contributions:** Z.Z.: Methodology, Validation, Review, Editing. S.F.: Methodology, Software, Validation, Writing—original draft. M.S.: Data curation, Conceptualization. Q.L.: Data curation, Investigation. All authors have read and agreed to the published version of the manuscript.

**Funding:** This work was funded by National Natural Science Foundation of China (No. 71971112); National Key R&D Program of China (2021YFB1600500).

**Data Availability Statement:** Not applicable.

**Acknowledgments:** This work was funded by National Natural Science Foundation of China (No. 71971112); National Key R&D Program of China (2021YFB1600500).

**Conflicts of Interest:** The authors declare no conflict of interest.

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
