# Peer review of "A Delay Prediction Method for the Whole Process of Transit Flight"

_aerospace, doi:10.3390/aerospace9110645_

Round 1

Reviewer 1 Report

A summary 

I see the main goal of the paper as the authors' efforts to expand knowledge and support solutions for delay prediction for the entire process of transit flights through the basic steps of node and connection time prediction and the identification of a delayed flight, within the Airport Collaborative Decision Making (ACDM) system.

The main contribution and strength of the paper are that it uses an experimental database of data from an airport with frequent traffic for model verification. The authors share knowledge from the analysis for the benefit of educating the aviation community about the issue, to achieve the goal "Action Plan for Promoting the Construction of Four Types of Airports (2020-2035)".

General concept comments

After getting acquainted with the abstract of the paper, it may not be apparent to the reader what research question the authors are solving.

Following the abstract of the paper, in the Introduction, a specific research question/hypothesis should be defined, which will be tested within the research of the issue. Finally, the main objective of the work is briefly stated and the main conclusions are highlighted.

In the Introduction: You clearly define the research question and highlight the main conclusions of your work. Subsequently, the structure of the paper in individual sections is presented.

The literature search on the issue is adequate.

The authors used and developed a relevant methodology for solving the problem.

The tables and figures correlate with the content of the paper, they are easy to interpret and understand (with various combinations of codes, but identifying stall-type accidents).

The references cited are correct in my opinion. The number of connections is modest, but it was enough for the author to solve the problem.

The paper has its limits.

The conclusions are formulated, by the presented evidence and arguments of the authors.

We can consider that the paper's focus can bring a new understanding of the need to create skills and competencies for airport management using the developed prediction model. The selected knowledge can be appropriately extracted for the airport environment of another country/region.

The paper has the potential to generate further research questions for continued scientific work. The final recognition rates of 83.6%, 90.4%, and 96.5% for delayed flights also confirmed the effectiveness and superiority of the method in this paper by comparing it with the results of existing studies.

The paper does not have a Conclusions section: but this section is not mandatory.

Specific comments 

In the Introduction

Identify the addressed research question/hypothesis in the wider context and purpose of the paper and highlight the main conclusions of your work.

I recommend the paper for publication as part of the academic and research discussion on the given topic, after minor revisions.

Reviewer 2 Report

Dear Authors, 

first of all, I would like to thank you for the opportunity to read your manuscript. I found it very interesting, some questions below:

* did you find any similar researches in other countries? I see only references from Chinese authors, a comparison with other countries/regions may give a broader view of the experiment carried out,

* did you have possibility to use "open available" data or is some kind of special database (you don't need to comment on this in your paper, just my curiosity)

* did you think about weather as one of the model parameters, 

I also suggest to a bit better describe used algorithms and maybe mention tools which has been used. 

Best wishes, 
